REGISTERED REPORT PROTOCOL

# Study protocol for COvid-19 Vascular sERvice (COVER) study: The impact of the COVID-19 pandemic on the provision, practice and outcomes of vascular surgery

**Ruth A. Benson[1,2☺]\*, Sandip Nandhra[3☺], on behalf of the Vascular and Endovascular Research Network[¶]**

1 Department of Cancer and Genomics, University of Birmingham, Birmingham, United Kingdom,
2 Department of Vascular Surgery, University Hospital Coventry and Warwickshire, Coventry, United Kingdom, 3 Newcastle University Northern Vascular Centre, Newcastle, United Kingdom

☺ These authors contributed equally to this work.
¶ Membership of the Vascular and Endovascular Research Network is listed in the Acknowledgments.
\* r.a.benson@bham.ac.uk

## Abstract

### Background

The novel Coronavirus Disease 2019 (COVID-19) pandemic is having a profound impact on global healthcare. Shortages in staff, operating theatre space and intensive care beds has led to a significant reduction in the provision of surgical care. Even vascular surgery, often insulated from resource scarcity due to its status as an urgent specialty, has limited capacity due to the pandemic. Furthermore, many vascular surgical patients are elderly with multiple comorbidities putting them at increased risk of COVID-19 and its complications. There is an urgent need to investigate the impact on patients presenting to vascular surgeons during the COVID-19 pandemic.

### Methods and analysis

The COvid-19 Vascular sERvice (COVER) study has been designed to investigate the worldwide impact of the COVID-19 pandemic on vascular surgery, at both service provision and individual patient level. COVER is running as a collaborative study through the Vascular and Endovascular Research Network (VERN), an independent, international vascular research collaborative with the support of numerous national and international organisations). The study has 3 'Tiers': Tier 1 is a survey of vascular surgeons to capture longitudinal changes to the provision of vascular services within their hospital; Tier 2 captures data on vascular and endovascular procedures performed during the pandemic; and Tier 3 will capture any deviations to patient management strategies from pre-pandemic best practice. Data submission and collection will be electronic using online survey tools (Tier 1: SurveyMonkey® for service provision data) and encrypted data capture forms (Tiers 2 and 3: RED-Cap® for patient level data). Tier 1 data will undergo real-time serial analysis to determine longitudinal changes in practice, with country-specific analyses also performed. The

**Data Availability Statement:** Data will be available upon the study's completion.

**Funding:** The study has received financial grant support from the Vascular Society of Great Britain and Ireland (VSGBI)/Circulation Foundation (no grant reference). The National Institute for Health Research (NIHR) has provided salary support for the co-chief investigators (reference: NIHR000359). The funders have had no role in the design, implementation, analysis or write up of the study.

**Competing interests:** The authors have declared that no competing interests exist.

analysis of Tier 2 and Tier 3 data will occur on completion of the study as per the pre-specified statistical analysis plan.

## Introduction

The novel coronavirus disease 2019 (COVID-19) pandemic is having an unprecedented effect on the provision of healthcare services worldwide. The delivery of surgical care to patients during this time is suffering as resources dwindle and hospital services are overwhelmed [1, 2]. It is essential to document the effect of this pandemic on the provision of vascular surgical services and the outcomes for vascular patients to guide future guidelines and provide foresight for potential problems beyond the pandemic.

Vascular patients are, for the majority, considered high risk for respiratory compromise and subsequent mortality if they contract COVID-19 [3, 4]. They are often frail, elderly, comorbid, and have less respiratory and physiological reserve than many others who contract the SARS-CoV2 virus [5]. A significant proportion of vascular surgical practice involves performing prophylactic operations to reduce the risk of a future cardiovascular event, for example carotid endarterectomy to prevent stroke or abdominal aortic aneurysm repair to prevent rupture. In addition, urgent or emergency surgical intervention to prevent the loss of limb or life, often through an endovascular and/or open revascularisation procedure, are commonplace.

Given the complex nature of vascular operations and the equally complex and co-morbid patient population, the COVID-19 pandemic presents a particularly challenging situation for the vascular surgeon. There is a delicate balance between the risk of a patient contracting or surviving COVID-19, the availability of critical care and anaesthetic support needed to perform high-risk vascular interventions, and the risk of limb loss, other significant morbidity or mortality for the patient from their presenting condition if treatment is unduly delayed.

A major curtailment of vascular practice has already occurred, with many vascular institutions postponing all but the most urgent surgery, choosing and endovascular surgical option where feasible, delaying routine clinic appointments, and using telephone consultations much more frequently [6]. Given the unparalleled nature of the situation, there is an urgent need to quantify the impact of COVID-19 on the provision of vascular surgical services, the adjustments made to vascular practice, and the consequence to patient care.

The COvid-19 Vascular sERvice (COVER) Study is a three-tiered study designed to capture global data on vascular practice(s) during the pandemic including how practice evolves over time, the effect on outcomes for patients presenting with, and/or receiving treatment for, vascular surgical conditions during the pandemic and in the subsequent months of global recovery.

### Study overview

COVER will be run as a worldwide collaborative research project. It will be led by the UK-based Vascular and Endovascular Research Network (VERN). VERN is an established independent, international vascular trainee research collaborative, which has previously designed and delivered several projects across the UK and internationally [7, 8].

The aims of the study are to understand and evaluate the impact of the COVID-19 pandemic on global vascular practice.

Centres and individuals will be invited to participate in the COVER study which will be advertised via VERN social media channels as well as via regional VERN representatives

(doctors, nurses and other healthcare professionals) and through our collaborative networks mailing lists. Engagement with each tier of the project by each collaborator is anticipated and outlined below.

## Materials and methods

The study has 3 'Tiers':

Tier 1 is a survey of vascular surgeons to capture longitudinal changes to the provision of vascular services within their hospital.

Tier 2 will capture data on vascular and endovascular procedures performed during the pandemic.

Tier 3 will capture management of referrals made to vascular teams, focusing on any deviations to patient management strategies from pre-pandemic best practice.

### Centre eligibility

All hospitals and networks which provide cover for elective and emergency vascular patients.

### Tier 1—Changes to unit-level clinical processes

**Primary objective.** To objectively capture the changes made to the structure and delivery of vascular surgery at unit level throughout the COVID-19 pandemic.

**Secondary objective.** International comparison of structural changes made to vascular practice.

**Outcomes.** *Primary Outcome.* Documentation of changes to structure and activities and processes within vascular services during the first peak of COVID-19 globally., including:

- Operations/interventions offered/not offered

- Thresholds for offering admission/intervention

- Seniority and/or number of specialists performing caseload

- Management of screening / surveillance programmes (AAA, post-EVAR, bypass graft surveillance, stent surveillance)

- Imaging availability

- Interventional radiology support and availability

- Management of multi-disciplinary team meetings

- Changes to trainee (resident/registrar) and consultant/attending rotas

- Outpatient clinic availability and format

- Use of vascular team members to cross-cover other specialties or clinical areas

- The availability of personal protective equipment (PPE)

*Secondary outcomes.* International comparison of changes made to vascular services, and documentation of how practice fluctuates as case numbers rise and fall using a surgeon derived scoring system.

## Materials and methods

The Tier 1 "service evaluation study" will be circulated to all interested centres and data collected via an online survey platform (SurveyMonkey®). This will be conducted upon each

centre registering to participate in the overall study bundle. This survey will be repeated at regular intervals to document ongoing changes to unit practice in response to changing circumstances. The intervals between the survey repeats will depend upon the progress of the pandemic. Collaborators will be updated regularly regarding survey outcomes. Responses will reflect unit practice as a whole and should therefore be a unified unit level response approved by the centre lead.

This information will be fed back to the relevant bodies (VSGBI, ESVS, SVS etc.) to allow real-time feedback on practicalities of updated guidelines. The information will also be circulated via social media.

International/continental comparisons will be performed, where possible, to describe relative change in practice. A score of 0 to 3 will be allocated to each answer based on perceived relative service reduction by 12 VERN healthcare-professionals ('0' representing no change,'3' representing most significant change; S2 File).

## Tier 2 –Vascular and endovascular procedural data capture

**Primary objective.** To capture data on all vascular and endovascular interventions being undertaken throughout the COVID-19 pandemic and report early and late outcomes.

**Secondary objective.** Comparison of short and long-term clinical outcomes for patients receiving a procedure during the COVID-19 pandemic and those operated on during time periods not impacted by the pandemic.

**Outcomes.** *Primary outcome.* Data demonstrating volume and types of vascular interventions performed during the pandemic. Specifically:

- Types of procedure performed

- Time taken from presentation to intervention

- Mode of referral (primary vs. secondary care)

- Site of surgery: hub or spoke hospital

- Imaging modalities used and timings

- UK National Confidential Enquiry into Patient Outcome and Death (NCEPOD) classification

- Operative technique(s) and device(s) used

- Mode(s) of anaesthesia

- Time to discharge

- Whether suspected or confirmed COVID-19 +ve at time of surgery, COVID-19 +ve after surgery, or COVID-19 -ve

- Documentation of changes to usual practice (type of procedure, type of anaesthetic, post-procedural destination)

*Secondary outcomes.* Outcome measures documented at 30 days, 6 and 12 months following intervention including:

- Re-admission

- Re-intervention

- All-cause mortality

- Disease-specific mortality

- Morbidity

- If COVID-19 +ve: respiratory outcome, admission to intensive care unit.

Any type of re-admission will be included during the follow-up in the clinical team consider it is linked to index procedure. Morbidity will be defined by condition specific complications and generic surgical and medical complications in the clinical report forms. A preliminary list of outcomes to be measured at six months and one year have been included in the (S1 File).

## Materials and methods

This will be undertaken for a 3-month period in the first instance. This time period is subject to change depending on how the pandemic progresses.

### Patient enrolment

Patients will be identified prospectively at the time of surgery. To ensure comprehensive data capture, patients may also be identified retrospectively following an emergency procedure for example. All patients receiving a vascular procedure are eligible for enrolment, including COVID-19 positive (+ve), COVID-19 suspected and COVID-19 negative (-ve). Inclusion and exclusion criteria will be specific to each country, due to the variation in approval processes internationally. However, broadly speaking:

*Inclusion criteria.* Any patient over the age of 18 undergoing an operation or procedure for a recognised vascular condition, including trauma. This includes patients who died on table during the procedure.

*Exclusion criteria.* Any procedure initially thought to require a vascular intervention which then did not e.g. laparotomy for suspected abdominal aortic aneurysm for which an alternative pathology was found. The requirement for patient consent for inclusion in the study will be approached on a centre by centre (or country by country basis).

### Tier 3 –Changes to acute vascular care management

**Primary objective.** To capture modification to the management of **all referred** urgent vascular cases during the COVID-19 pandemic and identify deviations from pre-pandemic best practice, standards and/or guidelines for acute/urgent cases due to healthcare pressures or resource limitations. This will focus on (but not be limited to) chronic limb-threatening ischaemia, symptomatic carotid disease, acute aortic syndromes and aortic aneurysmal disease.

**Secondary objective.** Linkage of data on change in practice to clinical outcomes at 6 and 12 months.

**Outcomes.** *Primary outcome.* Documentation of deviation from "best vascular practice" and the impact on patient care This will specifically focus on:

- Chronic Limb Threatening Ischaemia (CLTI) [9]:

  ○ Decision to discharge, admit or refer to an emergency ('hot') clinic

  ○ Decision for endovascular- or open surgical revascularisation first strategy

  ○ Decision for best medical therapy, palliation or primary amputation

- Symptomatic carotid disease:

- Patients managed with best medical therapy (BMT)

- Modifications to the indication and decision for carotid endarterectomy (CEA)

- Delays to treatment due to lack of resources, including operating theatre, anaesthetic support or bed availability

- Abdominal Aortic Aneurysm (AAA):

  - Use of endovascular repair +/- local anaesthesia

  - Changes to criteria for intervention

  - Decisions for palliation, i.e. 'turn down'

- Acute Aortic Syndrome (AAS):

  - Decision to manage in non-critical care beds

  - Changes to imaging protocol at unit level

  - Decision to defer intervention

*Secondary Outcomes.* Outcome measures at 6 and 12 months. Comparison of these outcomes with patients without change to management during the pandemic.).

- Example condition-specific outcome measures to include:

  - CLTI: limb salvage, amputation free survival, all-cause mortality

  - Carotid disease: ipsilateral stroke rate, any stroke rate, all-cause mortality

  - AAA: aneurysm-related mortality, all-cause mortality

- AAS: complication rate including rupture, all-cause mortalityOther vascular presentations such as via MDT, hot-foot clinic referrals.

Any type of re-admission will be included during the follow-up in the clinical team consider it is linked to index referral. Morbidity will be defined by condition specific complications and generic surgical and medical complications in the clinical report forms. A preliminary list of outcomes to be measured at six months and one year have been included in the (S1 File).

**Materials and methods.** This will take place over a minimum of one month and will invite vascular specialists to complete an anonymised proforma for every patient with any of the conditions listed above referred to the vascular service. Timings may change based on the duration of the pandemic.

**Patient enrolment.** Patients will be identified prospectively at the time of referral to the vascular team. To ensure comprehensive data capture, patients may also be identified retrospectively. All patients referred to vascular services are eligible for enrolment, including COVID-19 +ve, COVID-19 suspected and COVID-19 -ve. As for tier 2, inclusion and exclusion criteria will be specific to each country.

*Inclusion criteria.* Any patient over the age of 18 referred to the vascular team for a recognised vascular condition.

*Exclusion criteria.* Any patient initially thought to have a vascular condition requiring a vascular team review, which was subsequently found not to be a vascular condition e.g. a referral for abdominal pain? ruptured aortic aneurysm in a patient subsequently found to not have a AAA and another pathology instead. As with tier 2, the requirement for patient consent for inclusion in the study will be approached on a centre by centre (or country by country basis).

### Tier 2 and tier 3 data collection

Tier 2 and Tier 3 data (all anonymised and non-identifiable) will be collected and stored through a secure UK National Health Service server using the Research Electronic Data Capture (REDCap) web application. Designated collaborators at each participating site will be provided with REDCap project server login details, allowing them to securely submit data on to the REDCap system. REDCap has previously been successfully used for a range of other international cohort studies, including those led by GlobalSurg and the European Society of Coloproctology. The REDCap server is managed by the University of Birmingham, UK, with support provided by the GlobalSurg team.

Anonymised data will be collected relating to COVID-19 status, comorbidities, physiological state, treatment, operation or intervention, and outcome. A unique identifier will be assigned to each patient record. All participating centres will keep a record of patient details relating to the unique identifier for the collection of medium- and long-term outcome data and linking to the original participant record on REDCap

### Analyses

As this is a non-interventional study, analysis will be limited to presentation of numbers and proportions, with comparisons made to national and international standards. Interim analyses will be performed periodically to inform data collection and provide up to date information on the impact of the pandemic. The first formal analysis for Tier 2 will be performed once 50 patients have been entered onto the database, and the frequency of subsequent analyses will be determined by the findings of this. Hospital-level data will not be released or published by the VERN team, but individual centres will have full access to their own data.

### National and local approvals

Ethical approval from the UK Health Research Authority has been obtained for Tiers 2 and 3, permitting the capture of patient outcomes at 6 and 12 months (20/NW/0196 Liverpool Central, IRAS: 282224). The study is registered with ISRCTN registry (80453162).

Participating centres in the UK will be required to seek local research and development approval. Non-UK centres will need to obtain a research ethics committee or institutional review board approval in accordance with national and/or local requirements. The principal investigator at each participating site is responsible for obtaining necessary local approvals. Study organisational sponsorship is through the R&D Department at University Hospitals, Coventry and Warwickshire NHS Trust, Coventry, UK.

### Authorship

Collaborators from each site who contribute patients will be recognised on any resulting publications as PubMed-citable co-authors. The VERN model for collaborative authorship, that will be used for any disseminations arising from this project can be found here: https://vascular-research.net/authorship-policy/. An example of this can be found here: https://pubmed.ncbi.nlm.nih.gov/29452941).

## Discussion

The COVER study has been designed as the first vascular trainee led, multi-national prospective study of Vascular Surgical practice during the COVID-19 pandemic. It has several key points that will make it increasingly relevant in the current climate. A high mortality has been reported in elective general surgery patients who are COVID positive [10], which is

concerning for the vascular patient population who are at increased risk of succumbing to a COVID-19 infection owing to their older age, high levels of smoking and background respiratory conditions, and comorbidities including diabetes. These factors have all been linked to significantly reduced rates of survival in those that have contracted the SARS-CoV2 virus. Pre-existing conditions also mean that if our patients are admitted to hospital, they are less likely to be considered candidates for invasive ventilation due to the associated mortality reported [11]. In addition, COVID-19 associated coagulopathy is emerging as a presenting complaint for COVID-19 infection and will impact on vascular been observed; however it is unclear if this is a reduction in self-referrals to primary care due to fear of coming into hospital, or gatekeeping being performed by referring teams.

There is a familiarity within vascular surgery with the consequences of delays in presentation for key conditions such as acute aortic pathologies and CLTI, leading to fewer treatment options and poorer prognosis. As the pandemic progresses and elective operating is curtailed or stopped completely, there will be a growing list of patients who will require urgent surgery in the post-pandemic period once 'normal' service has resumed. This study will address the consequence of delaying surgery considered urgent or essential, and an understanding of the vascular caseload volume that is accumulating during the pandemic period that will need to be appropriately managed once the crisis has passed.

Through well-structured and purposeful collaborative working the VERN group developed and submitted the COVER study for ethical approval. This has been granted promptly to facilitate COVID-19 related research within vascular surgery. Similarly, the global vascular community has responded positively and over 150 centres have already participated in Tier 1 and registered interest for the other tiers across the globe indicating the support and global appeal of the study.

VERN and the COVER study has a strong trainee focus with the opportunity for trainees to contribute high quality data from their own centres. From previous work we have been able to demonstrate that trainees are highly motivated to participate in research when their efforts are recognised and PubMed citable [12]. This has been echoed by global collaborative studies such as those run by GlobalSurg [13] (https://globalsurg.org/), conducted under a single author name and listing all those individuals who have contributed as co-authors.

The protocol has been designed with the support of the Vascular Society of Great Britain and Ireland, who recognise the value and importance of accurate data collection during this period. The study data collection tools have been developed in close co-operation with our colleagues in Europe, the USA and Australia, to ensure questions are applicable and that possible answers reflect variations in practice around the globe. It has also helped to ensure that information is collected in sufficient depth to correlate with key outcomes at 6 and 12 months, without being onerous.

The study also benefits from the ongoing public dissemination of international vascular guidelines to support clinicians managing the current COVID-19 crisis. This will enable comparison of real time changes in practice against emerging and evolving guidelines. Furthermore, the availability of various national registries, that have benchmarked 'normal' practice and 'expected' condition-specific outcomes for key parameters against which there can be a detailed comparison of COVER study-reported practices and outcomes. Data points relating to patient, technical and peri-operative variables will also be compared between countries and wider regions to explore how differences in populations and practice may impact on disease specific outcomes at 6 and 12 months.

Tier 1 has already collected valuable real-time data which has been used to inform those who create and disseminate national guidelines [14]. Promoting inclusion amongst the global

vascular trainee community will also play a role in professional development, research skills and achievements which would not otherwise be available.

## Study limitations

COVER is a pragmatic real-world study. The nature of the current pandemic has meant that an appropriate sample size is difficult to calculate. Additionally, the dynamic nature of the COVID-19 pandemic means that numbers of cases, including those testing positive for or suspected to have infection with the SARS-CoV2 virus—hence a denominator for some calculations—are unknown. This insurmountable problem is not limited to this study and is currently frustrating efforts to determine appropriate strategies for lifting the lockdown in countries where the peak of the pandemic appears to have passed.

Many countries have scaled back elective practice and changed thresholds for operating on carotid stenosis and aortic aneurysms to varying degrees and along different timelines due to resource scarcities. This will have an impact on the volume of cases uploaded into Tier 2 and Tier 3 of the study, but the concurrent completion of Tier 1 should reflect this and has already provided live data from over 150 institutions in 45 countries across the globe. In the first instance, country-specific experiences have been shared to highlight practice changes around the globe with vascular colleagues. This information will also be used when analysing country-specific trends for surgery and referrals. This will allow inclusion of important variables such as type of hospital (private or government-run), loss of specialty firms with the redeployment of staff to support other specialties; the exact timing for milestones such as stopping screening programmes for aortic aneurysm, or moving to a practice of mainly best medical therapy for symptomatic carotid artery stenosis.

## Conclusion

Success of the COVER study will provide the global vascular community with robust data on the impact of the COVID-19 pandemic on our patients and the future legacy of delayed surgery and adjusted decision making. It will support further collaboration between vascular trainees globally, bringing together and recognising efforts to collaborate with colleagues around the world.

## Supporting information

**S1 File. Preliminary list of condition specific outcomes to be reported at 30 day, 6 and 12 month follow-up.**
(DOCX)

**S2 File. International/Continental comparison analysis.**
(DOCX)

## Acknowledgments

The Vascular and Endovascular Research Network executive committee: Ruth A Benson, Sandip Nandhra, Joseph Shalhoub, Athanasios Saratzis, David C Bosanquet, Rachael Forsythe, Sarah Onida, George Dovell, Louise Hitchman, Nikesh Dattani, Ryan Preece, Graeme K Ambler, Chris HE Imray.

The VERN executive committee would like to formally acknowledge: The research collaboration with the following groups: Vascupedia (European vascular education platform),

Australian and New Zealand Vascular Trials Network (ANZVTN), COVER Australia, Sing-Vasc (Singapore Vascular Surgical Collaborative).

The support of the following international groups: Vascular Society of Great Britain and Ireland (VSGBI), British Society of Endovascular Therapists (BSET), The Rouleaux Club, Audible Bleeding podcast, British Society of Interventional Radiology (BSIR), the BSIR trainees (BSIRT) and the European Society of Vascular Surgery (ESVS).

The department of research and development at the University Hospitals, Coventry and Warwickshire NHS Trust (Coventry UK). We would also like to thank Sonia Kandola, research business manager at the University Hospitals Coventry and Warwickshire NHS trust, who has been instrumental in establishing study sponsorship.

## Author Contributions

**Conceptualization:** Ruth A. Benson, Sandip Nandhra.

**Methodology:** Ruth A. Benson, Sandip Nandhra.

**Project administration:** Ruth A. Benson, Sandip Nandhra.

**Supervision:** Ruth A. Benson.

**Writing – original draft:** Ruth A. Benson.

**Writing – review & editing:** Ruth A. Benson, Sandip Nandhra.

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
