## [Decision Letter · Decision Letter 0]

28 Sep 2020

PONE-D-20-13638

Study protocol for COvid-19 Vascular sERvice(COVER) study : The impact of the COVID-19 pandemic on the provision, practice and outcomes of vascular surgery

PLOS ONE

Dear Dr. Benson,

Thank you for submitting your manuscript to PLOS ONE. After careful consideration, we feel that it has merit but does not fully meet PLOS ONE’s publication criteria as it currently stands. Therefore, we invite you to submit a revised version of the manuscript that addresses the points raised during the review process.

We look forward to receiving your revised manuscript.

Kind regards,

Francesco Di Gennaro

Academic Editor

PLOS ONE

Journal Requirements:

3. One of the noted authors is a group or consortium [Vascular and Endovascular Research

Network (VERN) Executive Committee]. In addition to naming the author group, please list the individual authors and affiliations within this group in the acknowledgments section of your manuscript. Please also indicate clearly a lead author for this group along with a contact email address.

Additional Editor Comments (if provided):

Dear authors, follow reviewer suggestion to improve your manuscript

Reviewers' comments:

Reviewer's Responses to Questions

**Comments to the Author**

1. Does the manuscript provide a valid rationale for the proposed study, with clearly identified and justified research questions?

Reviewer #1: Yes

Reviewer #2: Yes

2. Is the protocol technically sound and planned in a manner that will lead to a meaningful outcome and allow testing the stated hypotheses?

Reviewer #1: Partly

Reviewer #2: Yes

3. Is the methodology feasible and described in sufficient detail to allow the work to be replicable?

Reviewer #1: Yes

Reviewer #2: Yes

4. Have the authors described where all data underlying the findings will be made available when the study is complete?

Reviewer #1: Yes

Reviewer #2: Yes

5. Is the manuscript presented in an intelligible fashion and written in standard English?

Reviewer #1: Yes

Reviewer #2: Yes

6. Review Comments to the Author

You may also provide optional suggestions and comments to authors that they might find helpful in planning their study.

Reviewer #1: General comment

It is an original and timely study protocol. The proposed methodology is scientifically and technically sound. Minor revision with emphasis on methodological re-structuring is required. I have also attached a word document with specific comments. I hope the detail review, comments, and suggestions will help you to improve your protocol and to consider some unforeseen challenges during the implementation of your study proposal. Please, find the attachment.

Reviewer #2: It is an Interesting study of a 3-month period receiving vascular surgery. It is well written, structured, clear and easy to follow.In my opinion, It is a useful manuscrit with robust data. Overall, it is worth to read.

7. PLOS authors have the option to publish the peer review history of their article (what does this mean?). If published, this will include your full peer review and any attached files.

Reviewer #1: **Yes: **Serebe Gebrie

Reviewer #2: No

---

## [Author Response · Author response to Decision Letter 0]

16 Oct 2020

This has been changed – there is no data being generated directly from this paper. Any subsequent results from the study, if accepted by Plos One for publication, will be made fully accessible. 

3. One of the noted authors is a group or consortium [Vascular and Endovascular Research

Network (VERN) Executive Committee]. In addition to naming the author group, please list the individual authors and affiliations within this group in the acknowledgments section of your manuscript. Please also indicate clearly a lead author for this group along with a contact email address.

This has been changed and formatted as per your style requirements.

 Supporting material has been submitted with this application and referenced in the text.

This has been deleted from all sections other than the methods. 

Responses to reviewers

Comment to the Authors

Title: Study protocol for COvid-19 Vascular sERvice(COVER) study: The impact of the COVID-19 pandemic on the provision, practice, and outcomes of vascular surgery

General comment

 It is an original and timely study protocol. The proposed methodology is scientifically and technically sound. Minor revision with emphasis on methodological re-structuring is required. I hope the detail review comments and suggestions will help you to improve your protocol and to consider some unforeseen challenges during the implementation of your study proposal.

Thank you for your comments, we appreciate the time taken to read our protocol. We have responded to them and that the manuscript has been updated to reflect your suggestions. 

Specific comments

Abstract – Well summarized

Funding statement (L32) - It seems VERN is part of the Funding organization(VSGBI). 

If so, there may be a question of competing interest. You may need to clarify the difference between VERN and Vascular Society of Great Britain.

Mention of supporting organisations have been completely moved to the acknowledgements. We have included the following statements to clearly document the separation between the two entities:

1. Line 53, abstract: ‘an independent, international vascular research collaborative’

2. Line 119: ‘established independent, international’

3. In the funding statement: ‘The funders have had no role in the design, implementation, analysis or write up of the study.’

Method – Need to restructure the method section – I suggest having the ‘objective’ section before the method so that you can put the general and specific objectives. The objectives are fragmented and there are lots of redundancies. 

Although we have kept the overall structure similar, the objectives and outcomes are now at the start of each tier section, and redundant text has been cleared to reduce any repetition. We hope this is clearer for the reader.

L130 – 138, Move the paragraph about the “overview of VERN” into the introduction part. Probably at the end of the introduction and before method.

This has been moved

L139-141 – Move to the objective section or just put it at the end of the introduction. 

The study overview, with a brief description of the aims has been moved to the end of the introduction.

Strat method with L141 “ This study has 3 tiers ….”

Changed

L152 – Once you put the general and specific objectives for each tier above, you may not have to repeat the objectives for each tier. If you prefer to put under each tier, just avoid repetitions.

This has been edited to reflect the reviewer’s comments

L152 - I think if there is no secondary objectives or secondary outcomes, no need to level as ”Primary”. Apply this for the following tier as well.

The objectives have been clarified and broken down into clearer primary and secondary objectives

L166 -179 – Too many outcomes and some of these are activities that may define the listed outcomes. For example, conducting multi-disciplinary team meetings is an activity.

The primary outcome has been clarified to include ‘activities’. The team feel that for those departments who use the MDT, it is a separate activity to the other processes included in the primary outcome list. 

L185-189 – See comments above about objectives for each tier.

Addressed under comments above

L193-197, Patient enrolment

How about the exclusion criteria? Do you have any?

Details of inclusion and exclusion criteria have been added to the tier 2 and 3 sections to make it clearer for the reader.

Will you include unknown status for COVID-19? Refusals for testing?

Yes. COVID-19 status is not part of the inclusion or exclusion criteria. We are not documenting reasons for lack of testing formally. 

You are to include prospective patients. 

Yes, although to ensure comprehensive data capture, patients may also be identified retrospectively.

How do you minimize providers' bias? 

The role of principle investigator will be to ensure the protocol is followed, and that sequential patients are included in the study, in order to prevent this bias. This will be according to priciples documented in the declaration of Helsinki.

Will you let the providers/Vascular specialists know about the study?

Yes. Details of this were included in the materials and methods section, but have been moved up to the study overview section for clarity.

L216-224 – You may need to have an “Operational definition” section to define the outcome variables. For example, 

• How do you measure morbidity? Morbidity related to Vascular disease, related to intervention, COVID-19, or co-morbidities? 

Both will be measured. A full list of morbidity and mortality measures have been included in the supplementary material for reference. 

• How many days of sickness is considered as morbidity? Measure subjectively or objectively?

Number of days for a sickness won’t be included, just the record of a morbidity or complication. 

• The same thing does for Re-admission. 

As above, number of days is not a factor (we are not planning a cost analysis), only the fact that a re-admission occurred. 

• Within how many days after discharge is considered as re-admission? 

There is no time limit applied to re-admission, it will reflect its link to the index procedure of presenting complaint. 

• With the same case or re-admission for any case?

The following statement has been added to the sections for tiers 2 and 3:

“Any type of re-admission will be included during the follow-up in the clinical team consider it is linked to index [procedure (tier 2) OR referral (tier 3)]. Morbidity will be defined by condition specific complications and generic surgical and medical complications in the clinical report forms. A full list of outcomes to be measured at six months and one year have been included in the supplementary material. 

L234 – You will invite vascular specialists only to complete the survey. However, the burden of COVID-19 also affects other professionals like nurses, anesthesiologists, and surgical technologists. Why do you propose to include only vascular specialists in this study?

We assume this is in reference to Tier 1 – From our collaborative work, and communications with colleagues in other specialties we are aware of a multitude of other pieces of work designed to address the impact of the pandemic on our nursing, anaesthetic and allied medical colleagues. We do not want to repeat surveys already addressed by other specialties. Our focus is on the vascular community as we know it well, and hope to produce work that will help us all directly. 

Sampling and sample size determination - Although it is stated under limitation, it is better to explain how to include participants for both patients and vascular specialists. You can also put a sample size but justify it. 

Do you have any inclusion criteria for the Vascular Specialists?

The protocol has been updated with inclusion criteria for tiers 2 and 3.

How do you control/consider cofactors? 

This is a non-interventional study, and one of its’ nature has not been performed before, therefore we have no sample size. We will be describing and presenting results, and do not plan to perform any in depth control of cofactors at this stage. Once 50 patients have been entered we will review the data and assess if more details analysis is appropriate. 

Do you think all the situations, except related to COVID-19, are comparable pre- and during the pandemics? 

This has depended on the results of tier 1. Thus far, our results have shown that there have been major changes to how vascular surgery is practiced globally. This applies to patients with and without COVID-19. Therefore we believe it is reasonable to compare all factors of care occurring during the pandemic with those aspects during equivalent times in previous times.

Please put some assumptions that will justify your comparative analysis before and during the COVID-19 pandemics.

We have made no assumptions in this protocol, as this is an unprecedented situation. We will review our data after the first 50 patients to establish the assumptions we will make during analysis. We will also review data from other studies that are published in other specialties to establish relevant variables to include in outcome analysis.

References

• Some references missed the publication date or accessed date. Reference 1&12.

This has been updated

• I can't see the in-text citation for references 12&13. I am not sure but just check all the in-text citations appear in the reference lists and vice-versa.

This has been checked and updated

---

## [Decision Letter · Decision Letter 1]

19 Nov 2020

Study protocol for COvid-19 Vascular sERvice(COVER) study : The impact of the COVID-19 pandemic on the provision, practice and outcomes of vascular surgery

PONE-D-20-13638R1

Dear Dr. Benson,

We’re pleased to inform you that your manuscript has been judged scientifically suitable for publication and will be formally accepted for publication once it meets all outstanding technical requirements.

Kind regards,

Francesco Di Gennaro

Academic Editor

PLOS ONE

Additional Editor Comments (optional):

Dear Authors congratulations

Reviewers' comments:

Reviewer's Responses to Questions

**Comments to the Author**

1. Does the manuscript provide a valid rationale for the proposed study, with clearly identified and justified research questions?

Reviewer #1: Yes

Reviewer #2: Yes

2. Is the protocol technically sound and planned in a manner that will lead to a meaningful outcome and allow testing the stated hypotheses?

Reviewer #1: Yes

Reviewer #2: Yes

3. Is the methodology feasible and described in sufficient detail to allow the work to be replicable?

Reviewer #1: Yes

Reviewer #2: Yes

4. Have the authors described where all data underlying the findings will be made available when the study is complete?

Reviewer #1: Yes

Reviewer #2: Yes

5. Is the manuscript presented in an intelligible fashion and written in standard English?

Reviewer #1: Yes

Reviewer #2: Yes

6. Review Comments to the Author

You may also provide optional suggestions and comments to authors that they might find helpful in planning their study.

Reviewer #1: All my comments and concerns are well addressed. Better clarifications are provided for those clarity issues. Overall the protocol is significantly improved. I feel this manuscript is scientifically and technically sound for publication.

Reviewer #2: Changes made after comments in the second version improved the manuscript quality, spcecially the methodology section. I would definetely recommend the new version for publication.

7. PLOS authors have the option to publish the peer review history of their article (what does this mean?). If published, this will include your full peer review and any attached files.

Reviewer #1: **Yes: **Serebe Gebrie

Reviewer #2: **Yes: **Mireia Alcalde

---

## [Editor Report · Acceptance letter]

18 Dec 2020

PONE-D-20-13638R1 

Study protocol for COvid-19 Vascular sERvice (COVER) study: The impact of the COVID-19 pandemic on the provision, practice and outcomes of vascular surgery 

Dear Dr. Benson:

I'm pleased to inform you that your manuscript has been deemed suitable for publication in PLOS ONE. Congratulations! Your manuscript is now with our production department. 

Kind regards, 

on behalf of

Dr. Francesco Di Gennaro 

Academic Editor

PLOS ONE